# QSTR Modeling to Find Relevant DFT Descriptors Related to the Toxicity of Carbamates

**DOI:** 10.3390/molecules27175530

**Published:** 2022-08-28

**Authors:** Emma H. Acosta-Jiménez, Luis A. Zárate-Hernández, Rosa L. Camacho-Mendoza, Simplicio González-Montiel, José G. Alvarado-Rodríguez, Carlos Z. Gómez-Castro, Miriam Pescador-Rojas, Amilcar Meneses-Viveros, Julián Cruz-Borbolla

**Affiliations:** 1Área Académica de Química, Centro de Investigaciones Químicas, Universidad Autónoma del Estado de Hidalgo, km. 4.5 Carretera Pachuca-Tulancingo, Ciudad del Conocimiento, Mineral de la Reforma 42184, Mexico; 2Escuela Superior de Cómputo, Instituto Politécnico Nacional, Yautepec 62739, Mexico; 3Departamento de Computación, CINVESTAV-IPN, Av. IPN 2508, Col. San Pedro Zacatenco, Ciudad de México 07360, Mexico

**Keywords:** QSTR, toxicity, carbamates, DFT, acetylcholinesterase

## Abstract

Compounds containing carbamate moieties and their derivatives can generate serious public health threats and environmental problems due their high potential toxicity. In this study, a quantitative structure–toxicity relationship (QSTR) model has been developed by using one hundred seventy-eight carbamate derivatives whose toxicities in rats (oral administration) have been evaluated. The QSRT model was rigorously validated by using either tested or untested compounds falling within the applicability domain of the model. A structure-based evaluation by docking from a series of carbamates with acetylcholinesterase (AChE) was carried out. The toxicity of carbamates was predicted using physicochemical, structural, and quantum molecular descriptors employing a DFT approach. A statistical treatment was developed; the QSRT model showed a determination coefficient (*R*^2^) and a leave-one-out coefficient (*Q*^2^*_LOO_*) of 0.6584 and 0.6289, respectively.

## 1. Introduction

For decades, pesticides have been widely employed to either prevent, destroy, attract, repel, or control unwanted pests in plant or animal species [1]. There are different types of pesticides considering their chemical structures, e.g., organochlorides, organophosphates, carbamates, and pyrethroids, among others [2]. Particularly, carbamate compounds, derived from carbamic acid, have been used as pesticides to provide broad-spectrum control of insects around the world due to their broad biological activity and low bioaccumulation [3,4,5]. It is known that the residues from carbamates can interact with the human body through the food chain; due to their wide use to prevent, control, or eliminate diseases, insects, and grasses that are harmful in agricultural production, carbamate compounds could pose a threat to human health [6,7,8]. In this vein, to evaluate the toxicity of chemical compounds, the most common method is based on the lethal dose at 50% (*LD*_50_), which is the minimal dose that causes death in 50% of individuals in a sample [2]. 

From a mechanistic view, several studies have shown that the toxicity of carbamate compounds is mediated by the inhibition of the enzyme acetylcholinesterase (AChE), where the first step in the inhibition process involves the formation of an enzyme–inhibitor complex with its subsequent carbamylation by the serine-hydroxyl group, generating a carbamate on the serine residue that is no longer able to hydrolyze the acetylcholine substrate [9].

Using these data, a quantitative structure–toxicity relationships (QSTR) analysis can be used to generate a mathematical model to predict the toxicity of new or unassessed compounds such as carbamates that are structurally related to the model training set [10,11]. 

Thus, we describe herein, that the analysis of the chemical reactivity of carbamates can generate a reliable model to relate their local and global DFT descriptors with their toxicity via a QSTR study. The QSTR model obtained, describes a relationship between molecular descriptors and toxicity. The methodology presented here is based on the experimental toxicity from a set of carbamates and the subsequent calculation of their molecular descriptors to model the relationship of physicochemical or structural properties with toxicity.

## 2. Results and Discussion

### 2.1. Structure and Optimization

A set of one hundred seventy-eight carbamate-based compounds (Figure 1 and Appendix A) with *LD*_50_ on rats reported in the ChemID database, were optimized by utilizing DFT with the exchange–correlation functional PBE and the 6-311+G* basis set. The optimized structures of all carbamates used in the present study are shown in Appendix A.

### 2.2. Electronic Analysis and DFT Descriptors of Carbamates

A frontier molecular orbital analysis was performed to determine the differences in the reactivity of one hundred seventy-eight carbamates. The chemical structures and graphs of the HOMO and LUMO frontier molecular orbitals were studied, and some examples are displayed in Figure 2; the full set of frontier molecular orbitals is shown in Appendix A. Frontier molecular orbital analysis of all carbamates revealed that HOMO orbital is mainly located in the aromatic ring with a small contribution towards the adjacent oxygen and nitrogen atoms of the carbamoyl group; on the other hand, when the carbamates lacks of an aromatic ring, the HOMO was mainly located on the oxygen and the adjacent atoms of the carbonyl group of the carbamate moiety, while the LUMO orbital is mainly located on the conjugated carbon atoms and on the carbon atom of the carbamate moiety, which suggests that these sites are labile and allow a nucleophilic attack by the AChE enzyme [1].

To evaluate the relationship between the chemical reactivity and toxicity of the carbamates, DFT calculations were performed. The values of the electron affinity (*A*), ionization potential (*I*), chemical potential (*μ*), hardness (*η*), electrophilicity (*ω*), and Hirshfeld charges of the carbon atom in the carbonyl group (*qC*) are reported in Table 1. All descriptors were calculated using the PBE/6-311+G* level of theory; the results showed that the HOMO energies (*E*_HOMO_) have values from –1.23 to 1.83 eV, and LUMO energies (*E*_LUMO_) from 6.05 to 9.96 eV, generating a mean gap (Δ*E*) of 8.08 eV with a standard deviation of 0.81 eV. 

### 2.3. QSTR Modeling

By applying a genetic algorithm over selected reliable molecular descriptors, a model with ten descriptors was obtained. The electronic affinity (*EA*) was selected as the first descriptor; this property is a global electronic descriptor, and it is associated with the LUMO energy, while *qC* was selected as the second descriptor, which is an electronic local descriptor related to the Hirshfeld charge of the carbonylic carbon atom.

Furthermore, another eight structural parameters were obtained in the model; these descriptors such as the lopping centric index (*LOC*) [12], the normalized spectral positive sum from reciprocal squared geometrical matrix (*SpPosA_RG*), the H autocorrelation of lag 4 weighted by mass [13] (*H4m*), the number of total tertiary C(sp^3^) (*nCt*), the presence (value of 1) or absence (value of 0) of an aliphatic substituent bonded to the sp^3^ oxygen of the carbamate group, (*nROCON*; for example, in Figure 3 there is an example of a value of 1 for this parameter), the presence (value of 1) or absence (value of 0) of C-N and N-O at topological distance of five bonds respectively (*B05*_[C-N]_ and *B05*_[N-O]_; see Figure 3), and, lastly, *DLS*_05_, that is the fifth drug-like score based on the two rules proposed by Zheng et al. [14], where the first rule is *n*NO/*n*C3 in the range 0.10–1.80 which is an index related to the proportion of heteroatoms, defined as the ratio of the total number of oxygen and nitrogen atoms (*n*NO) over the number of carbon atoms with sp^3^ hybridization (*n*C3) and the second rule is *Unsat*-*p* ≤ 0.43, where *Unsat*-*p* is a measure of molecule unsaturation and is defined as is the ratio of molecular unsaturation, as defined by the *Unsat* index, over the number of atoms which do not have bonded hydrogens and halogens. The *Unsat* index [14] is calculated from Equation (1) where NRG_567_ is the number of 5-, 6-, and 7-membered rings, nDB the number of double bonds, nTB the number of triple bonds, and nAB the number of aromatic bonds.
*Unsat* = NRG_567_ + nDB + 2⋅nTB + (nAB + 1)/2(1)

The best obtained model (Table 2) was selected considering the criteria for predictive models proposed by Golbraikh et al. [15], with an *R*^2^ = 0.6584 and a *Q*^2^*_LOO_* = 0.6289. The result was obtained using the ten above-described parameters. The data used to generate the model is displayed in Appendix A, and the correlation matrix for these descriptors is shown in Table 3. The descriptors showed a low correlation with each other; this implies that all variables are important in the multi-linear regression model (MLR). Additionally, a five-fold cross-validation was performed and the mean values for R squared, the root mean square error (RMSE), and the mean absolute error (MAE) were 0.6442, 0.4855 and 0.3889 respectively, this result confirms the proposed model has a good selection of test and training set, full information of cross-validation models is shown in Appendix A. Furthermore, we test several regression approaches such as Ridge, Lasso, Backward-Forward selection, XGBoost and support vector regression (SVR) with the score *R*^2^ in a range of 0.67 to 0.88; although SVR gives a better score it doesn’t allow a chemical interpretation of the model.

The *R*^2^ value denotes whether the model obtained is viable or not; i.e., if the linear association between the toxicity [quantified by log(1/C)] and the molecular descriptor in the model is strong enough. Thus, as the value of *R*^2^ approaches unity, then the model is considered adequate. Dispersion plots to validate the condition of the linear relation between descriptors and predicted log(1/C) are displayed in Figure 4. All variables are distributed randomly around axis X. Also, in Figure 4 the horizontal and vertical axes indicate the experimental values and the toxicity values obtained for the carbamate compounds, respectively. The graph with leave–one–out (LOO) validation (Figure 5) shows that the compounds maintain the same trend as in the main model, i.e., quantitatively, the difference *R*^2^ − *Q*^2^ equal to 0.0295 does not exceed the range of values between 0.2 and 0.3, ref. [16] therefore this model is considered as acceptable.

The analysis of some statistical parameters (Table 2) shows that the variable that most contributes to the model is the *nROCON* electronic property, with a standardized coefficient of −0.3527 and the *p*-value equal to 1.308 × 10^−6^ which does not exceed the critical value of 0.05 [17]; a similar effect was observed on the other structural descriptors, therefore all variables that integrate the model are accepted.

Visualizing the applicability domain (AD) of the model and demonstrating the relationship between standardized residuals and leverage values (*h*), a Williams plot was utilized (Figure 6). The Williams plot displays all the data points that are surrounded by standard deviations (−3.0 σ, 3.0 σ) and most of them are behind the critical leverage value (h*) for the model, revealing the robustness of the AD for the present QSAR models, corroborating the compounds from the internal validation set remain within the applicability domain, with a leverage value lower than h* of 0.217. It should be noted that one compound that is part of the test set has a value closest to the 3.0 σ limit; however, this is not considered as atypical data, ref. [9] confirming that all compounds from the internal validation set remain inside the limit permissible error.

### 2.4. Binding of Carbamate Compounds to the Catalytic Site of AChE

Molecular docking calculations were performed to generate plausible carbamate-AChE complex models and evaluate their ability to reach the catalytic triad at the enzyme’s active site. The binding of a carbamate compound acting as an electrophile to the catalytic pocket of the AChE in suitable orientation is considered key to the inhibition of the enzyme by this family of compounds [18]. The inactivation of the enzyme depends upon the chemical reaction among the enzyme residue Ser200 to the carbonyl carbon atom of the carbamate ligands. From the generated models, it was observed that most of the tested compounds achieved proximity to the Ser200 ligand with a suitable orientation to allow a nucleophilic attack from the Ser200 hydroxyl group to the carbamate carbonyl group in each ligand. Figure 7 shows an example of this binding for the compound with Id 0000050077 (in purple). The hydrogen bond network formed between Ser200, His440, and Glu327 (highlighted in green), facilitates the subtraction of the hydroxyl H atom from Ser200, thus favoring the reaction of the Ser200 oxygen atom to the carbamate’s carbonyl group located, in this case, at just 2.82 Å distance. Most of the compounds showed distances within 6.0 Å, confirming their potential to inactivate the enzyme by chemical reaction with the catalytic serine residue.

Unfortunately, we could not find any direct relationship between the affinity or the distance to the catalytic center with the toxicity of the studied compounds. Therefore, it is concluded that the binding process would not be the most relevant stage in the inactivation mechanism of AChE by the carbamates herein studied. It is very likely that the reactivity or the activation barrier during the reaction could better explain the toxicity of these compounds.

## 3. Materials and Methods

### 3.1. Data Collection and Electronic Descriptors

All carbamate-based compounds were obtained from the ChemID database [19]. The *LD*_50_ values (mg/kg) (Appendix A) for all compounds were obtained under similar experimental conditions (oral administration in rats). For modeling purposes, *LD*_50_ values were converted to molar concentration in mmol/kg (C) and subsequently to negative logarithms (log 1/C) according to the literature [20,21].

3D molecular structures for all carbamate compounds were built using the reported structure from the ChemID database [19]. After, the structures were optimized using DFT, without symmetry constraints by using the Gaussian 09 program [22] with the functional PBE [23] in combination with the 6-311+G* basis set for all atoms. Additionally, the global electronic descriptors such as the energy level of the highest occupied molecular orbital (*E*_HOMO_), the energy level of the lowest unoccupied molecular orbital (*E*_LUMO_), chemical potential (*μ*), hardness (*η*), and electrophilicity (*ω*) for all carbamates were obtained based on DFT calculations [24,25].

Furthermore, local electronic descriptors were calculated for all carbamate groups from each molecule; these calculations were performed according to Vela et al. study [26] using Multiwfn [27].

### 3.2. Dragon Molecular Descriptors

Different types of molecular and electronic descriptors were used to develop the QSTR models. The Dragon software [28] was used to obtain the molecular descriptors based on the optimized molecules. Several non-representative descriptors (e.g., those showing the same values for all the compounds) were excluded from the data set. For each pair, if the correlation coefficient was higher than ~90%, then the pair was considered inter-correlated and one of both was excluded. The most significant descriptors that enabled the toxicity correlation were isolated from the data set and combined with the electronic descriptors in order to develop a good QSTR model.

### 3.3. QSTR Model Building and Validation

The entire set was split using a random selection of variables, using 15% of the full set as a test set. Potential QSTR models were obtained with the combination of global, local, and structural descriptors using QSARINS software [29], where the reliability of the QSTR model was internally validated by using the coefficient of determination according to Equation (2), where *y_obs_* and
${\bar{y}}_{obs}$ are the experimental log(1/C) and their average, respectively, while *y_calc_* is the calculated log(1/C) obtained via the QSTR model. The statistical analysis was performed with R-4.1.0 software project, available online: http://www.R-project.org/ (accessed on 1 June 2022). The multi-linear regression model was computed using the function lm (fitting linear models), which employs the QR factorization method to solve linear least-squares problems.
(2)$$R^{2\;}=1-\frac{\Sigma {\left(y_{obs\;}-y_{calc}\right)}^{2}}{\Sigma {\left(y_{obs}-{\bar{y}}_{obs}\right)}^{2}}$$

The data were divided into two sets: training and test sets using cross-validation and selecting the groups by the median of *R*^2^ value. The validation was performed using the leave-one-out coefficient, *Q*^2^*_LOO_*, in Equation (3), an internal validation that enables the predictability of the model to be ascertained, wherein, if *Q*^2^*_LOO_* > 0.5 considering the molecules from the set used to build the model, then the QSTR model is acceptable. This is achieved by calculating the log(1/C) of a molecule via a model that excludes the molecule which is cycled over the whole set of molecules.
(3)$$Q_{LOO}^{2}=1-\frac{\Sigma {\left(y_{obs\;}-y_{pred,\;LOO}\right)}^{2}}{\Sigma {\left(y_{obs}-{\bar{y}}_{obs}\right)}^{2}}$$

Moreover, the applicability domain (AD) was determined through leverage calculation, wherein all the untested compounds corresponding to the AD of the model will be predictable, while those compounds that do not fall within the AD are extrapolations obtained by the model [30]. In addition, to visualize the AD, a Williams plot was generated for the graphical detection of atypical responses (atypical Y) and the identification of the influence of the compounds in the model (atypical X).

### 3.4. Molecular Docking Calculations

All the studied carbamate compounds were docked into the active site of the target acetylcholinesterase (AChE) enzyme using the Autodock 4.2 program [31]. The AChE PDB model 2WFZ was selected to perform the calculations because it shows high resolution (1.95 Å) and good validation metrics [32]. This model was crystallized in a non-aged state with the organophosphate agent Soman, bonded to the catalytic residue Ser200. All the co-crystallized non-protein ligands were removed from the model and the remaining protein structure was subjected to a standard preparation workflow. Missing atoms in the protein were restored and polar hydrogen atoms were added to be treated explicitly in the calculations. Kollman charges and solvation parameters were assigned, and the structure was used to generate atom affinity maps for the calculation employing cubic grids of 90 × 90 × 90 points separated by 0.375 Å, centered at the enzyme’s active site. For ligand structures, only polar hydrogens were considered with Gasteiger atom charges and full ligand flexibility. The docking calculation consisted of 100 runs per ligand of the genetic-Lamarkian algorithm, using 150 individuals as the initial population, 2,500,000 energy evaluations, and 27,000 maximum number of generations. The resulting 100 models per ligand were clustered with a 2 Å cutoff for further analysis. The selected models for each complex were evaluated for their ability to reach the catalytic triad located at the bottom of the binding pocket of the enzyme, i.e., Ser200, His440, and Glu327.

## 4. Conclusions

The generated QSTR model to predict the toxicity of carbamate compounds contains ten descriptors, where two of them are DFT descriptors, the Hirshfeld charge of the carbon from carbonyl (*qC*) and the electronic affinity (*EA*), which suggests that the DFT parameters play an important role to determine the toxicity in this new model. A comparison of DFT descriptors suggested the best contribution for the model is promoted by the *EA*, which is associated directly with the interaction of carbamate compounds with its AChE enzyme target according to molecular docking calculations, where a suitable orientation and proximity of the carbamate moiety of most compounds is effectively achieved to favor reactivity with the enzyme’s catalytic residues via a nucleophilic attack. On the other hand, the *nROCON* descriptor generates the highest contribution in the model and indicates that a carbamate compound would be more toxic when an aromatic fragment is bonded to the sp^3^ oxygen atom from the carbamate group, and thus, this feature would favor the electrophilic character of the carbamate’s carbonyl group, further enhancing the inactivation reaction of AChE.

## Figures and Tables

**Figure 1 molecules-27-05530-f001:**
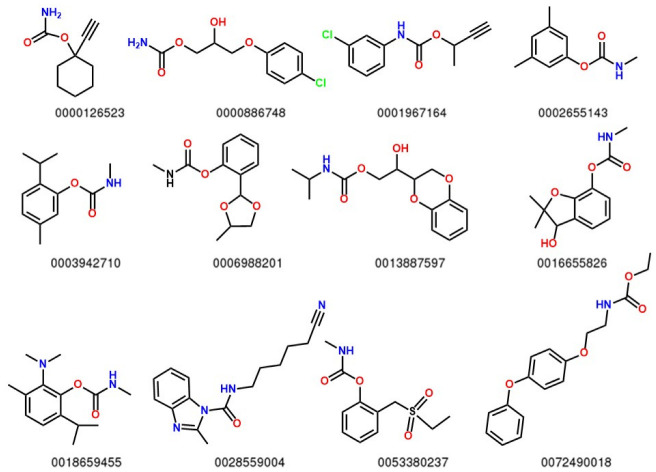
Twelve molecules randomly selected to represent the entire set of carbamates; the numbers are the ids from ChemID (Full set is reported in Appendix A).

**Figure 2 molecules-27-05530-f002:**
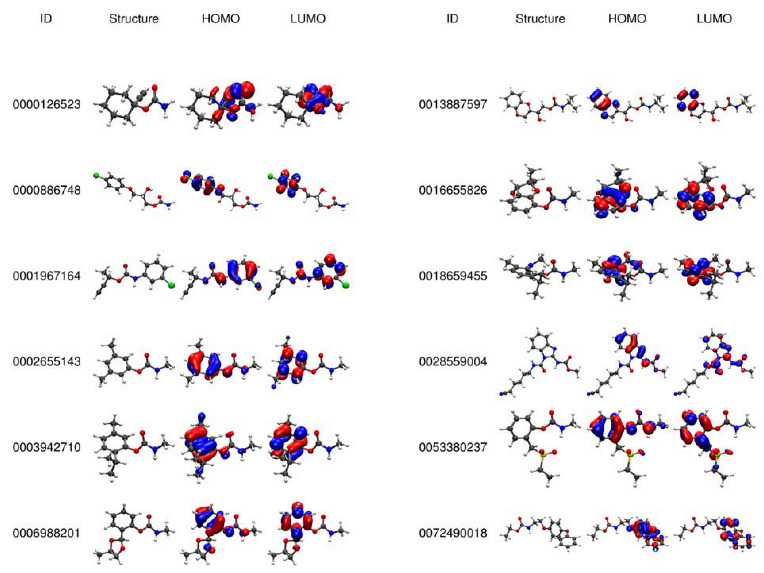
3D structure and frontier molecular orbitals of the twelve molecules selected randomly (full set is reported in Appendix A).

**Figure 3 molecules-27-05530-f003:**
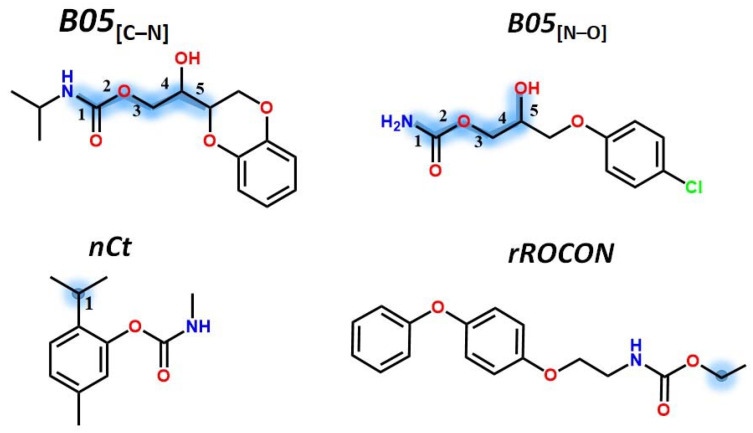
Variables are illustrated using some compounds of the set and marked with blue on the structure.

**Figure 4 molecules-27-05530-f004:**
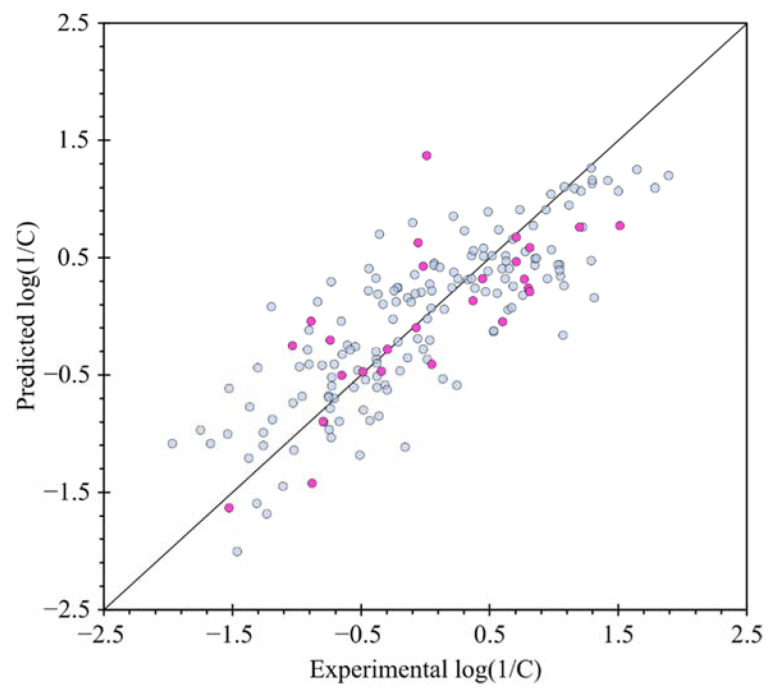
Graph of predicted values of log(1/C) *vs.* experimental values of log(1/C) for training set (light blue dots) and for test set (violet dots); solid line is a diagonal plot.

**Figure 5 molecules-27-05530-f005:**
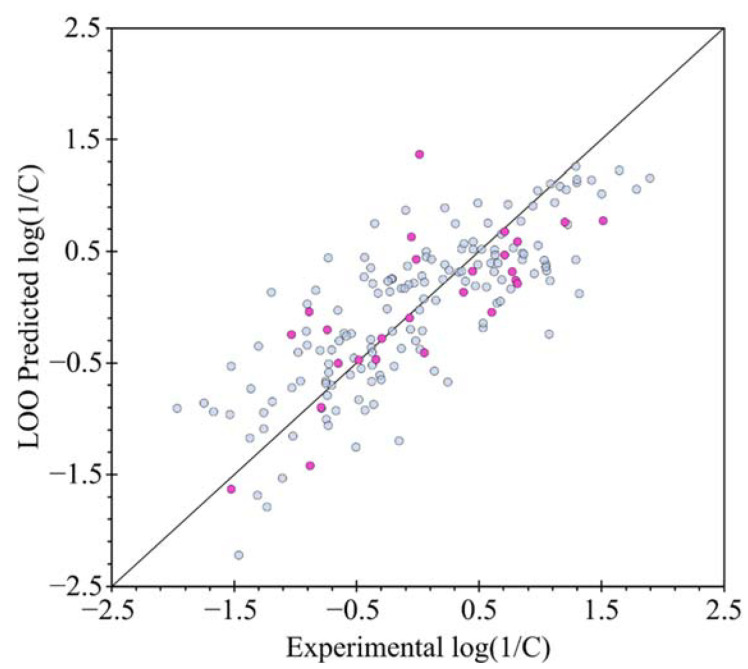
Graph of predicted LOO values of log(1/C) *vs.* experimental values of log(1/C) for training set (light blue dots) and for test set (violet dots), solid line is a diagonal plot.

**Figure 6 molecules-27-05530-f006:**
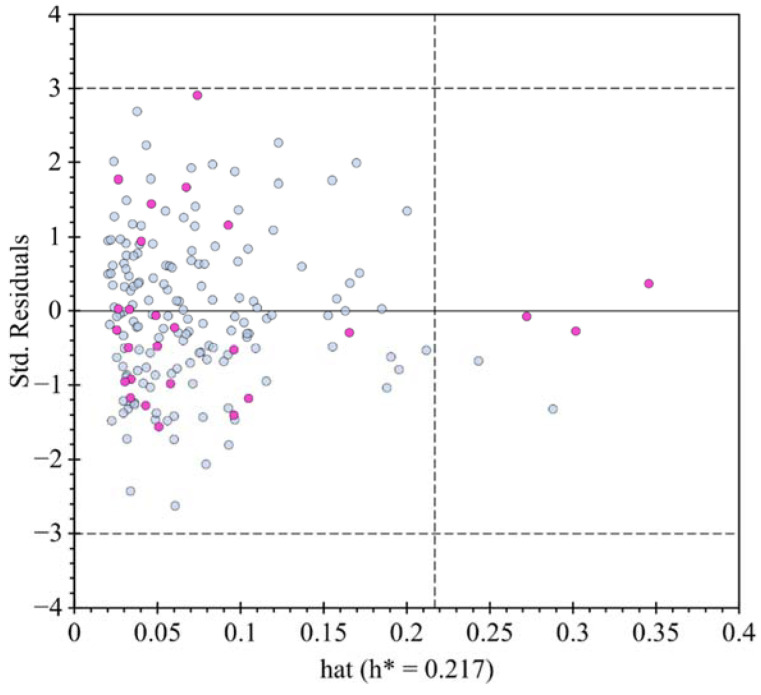
William’s plot with an up and low limit of 3σ and −3σ respectively, for the training set (light blue dots) and for the test set (violet dots).

**Figure 7 molecules-27-05530-f007:**
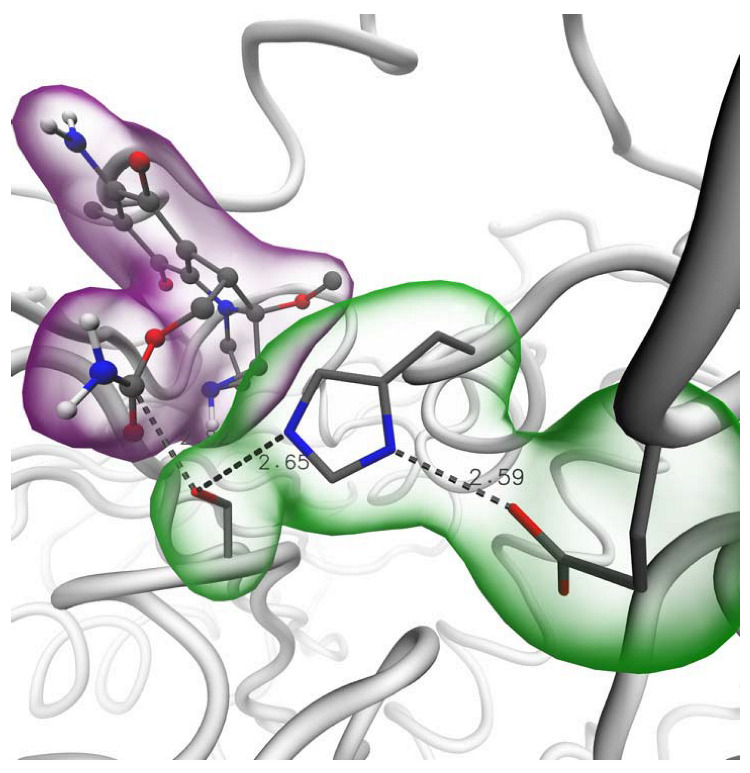
Model complex of Mitomycin approaching the catalytic triad in AChE (Ser200, His440, Glu327, in green).

**Table 1 molecules-27-05530-t001:** Model data for twelve compounds randomly selected; *EA* is in eV and *qC* in C, the other quantities are dimensionless.

*ID*	*Set*	log(1/C)	*EA*	*qC*	*LOC*	*SpPosA_RG*	*H4m*	*nCt*	*nROCON*	*B05* _[C-N]_	*B05* _[N-O]_	*DLS_05*
0000126523	Test	−0.29657	−8.643	0.1862	1.528	0.438	0.091	1	1	1	0	1
0000886748	Training	−0.48124	−7.7613	0.1931	2.25	0.424	0.26	0	1	1	1	1
0001967164	Training	−1.02699	−7.9976	0.198	2.071	0.415	0.257	0	0	1	0	0.5
0002655143	Training	−0.13579	−7.9584	0.1875	1.662	0.43	0.06	0	0	1	0	1
0003942710	Training	−0.38243	−7.8397	0.1869	1.697	0.432	0.093	1	0	1	0	1
0006988201	Training	0.33382	−7.938	0.189	1.403	0.425	0.112	0	0	1	0	1
0013887597	Test	−0.72691	−7.4242	0.1839	1.977	0.433	0.335	0	1	1	1	1
0016655826	Training	1.11994	−7.6752	0.186	1.481	0.431	0.107	1	0	1	1	1
0018659455	Training	0.57246	−7.084	0.1917	1.656	0.431	0.169	1	0	1	1	1
0028559004	Training	−0.88027	−7.7277	0.1988	2.473	0.429	0.371	0	0	1	0	0.5
0053380237	Training	−0.2937	−7.8825	0.1937	1.849	0.417	0.138	0	0	1	0	0.5

**Table 2 molecules-27-05530-t002:** Best obtained model, where G, L, and S means global, local, and structural descriptors respectively, Coeff mean coefficient, Std. Coeff is the standardized coefficient and Co. Int. is the confidence interval at 95%.

Variable	Type	Coeff.	Std. Coeff.	Co. Int.
*EA*	G	0.3231	0.2267	0.2055
*qC*	L	−34.0837	−0.1732	23.1746
*LOC*	S	−0.6319	−0.2946	0.2317
*SpPosA_RG*	S	−22.6053	−0.2935	8.3602
*H4m*	S	−1.5012	−0.3124	0.6143
*nCt*	S	0.2275	0.1312	0.1733
*nROCON*	S	−0.6919	−0.3527	0.2705
*B05* _[C-N]_	S	0.6524	0.1764	0.374
*B05* _[N-O]_	S	0.3996	0.2365	0.1956
*DLS* _5_	S	0.6244	0.2671	0.2492
*Intercept*	-	18.7033	-	6.3069

**Table 3 molecules-27-05530-t003:** Correlation matrix between variables of the model.

	*EA*	*qC*	*LOC*	*SpPosA_RG*	*H4m*	*nCt*	*nROCON*	*B05* _[C-N]_	*B05* _[N-O]_	*DLS_05*
** *EA* **	1.00									
** *qC* **	0.12	1.00								
** *LOC* **	−0.31	0.17	1.00							
** *SpPosA_RG* **	−0.09	−0.32	0.05	1.00						
** *H4m* **	0.44	0.09	−0.22	−0.16	1.00					
** *nCt* **	−0.08	−0.09	−0.05	0.19	0.02	1.00				
** *nROCON* **	−0.37	−0.45	0.14	0.25	0.19	0.08	1.00			
** *B05* ** ** _[C-N]_ **	0.32	−0.07	−0.05	0.00	0.14	0.12	−0.12	1.00		
** *B05* ** ** _[N-O]_ **	0.43	−0.13	−0.16	0.01	0.43	−0.07	0.16	0.03	1.00	
** *DLS_05* **	0.07	−0.25	0.05	0.33	−0.04	0.24	0.14	0.14	0.03	1.00

## Data Availability

All data generated or analyzed during this study are included in this published article.

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
