# Peer review of "QSTR Modeling to Find Relevant DFT Descriptors Related to the Toxicity of Carbamates"

_molecules, 2022, doi:10.3390/molecules27175530_

Round 1
Reviewer 1 Report
1. The performance of this model is too poor. The authors used multi-linear regression model, they can try different algorithm or parameters.
2. Why did the authors convert LD50 values to log 1/C? And how?
3. The authors should provide the results of cross-validation.
4. I suggest that the authors expose the code so that the reader can reproduce the results
Reviewer 2 Report
The toxicity of carbamates was predicted using physicochemical, structural, and quantum molecular descriptors employing a DFT approach. A statistical treatment was developed.
The conclusions are not clear and decisive. According to the authors, the following is stated:
The toxicity of carbamates can be estimated by a mixed QSTR model containing six descriptors, where one of them is a DFT descriptor while the other five are Dragon descriptors. The Hirshfeld charge and EA were a good DFT model descriptors due it has the best contribution in the model allowing the prediction of the toxicity and these are associated directly with the enzyme interaction according to the molecular docking calculations.
According to nROCON coefficient, the toxicity is directly associated with the kind of bond, if the sp3 oxygen is linked with an aromatic fragment is more toxic than an aliphatic one.
These last two paragraphs should be explained better, because it does not give a negative or positive answer. In general, the conclusions are erratic and perhaps the authors should design another theoretical procedure to support their studies.
Round 2
Reviewer 1 Report
1. The authors said they tested orther regression algorithms such as the Lasso and Ridge. They should provide all results about these test.
2. I suggest the authors try XGBoost or SVM.
3. is there a new version of supplementary file? I can only download the old version of supplementary file
4
Reviewer 2 Report
I believe the manuscript has been sufficiently improved to warrant publication in Molecules.
